# Amelioration of Nicotine-Induced Conditioned Place Preference Behaviors in Mice by an FABP3 Inhibitor

**DOI:** 10.3390/ijms24076644

**Published:** 2023-04-02

**Authors:** Wenbin Jia, Ichiro Kawahata, An Cheng, Takuya Sasaki, Toshikuni Sasaoka, Kohji Fukunaga

**Affiliations:** 1Department of Pharmacology, Graduate School of Pharmaceutical Sciences, Tohoku University, Sendai 980-8578, Japan; 2Department of CNS Drug Innovation, Graduate School of Pharmaceutical Sciences, Tohoku University, Sendai 980-8578, Japan; 3Department of Comparative and Experimental Medicine, Brain Research Institute, Niigata University, Niigata 951-8585, Japan; 4BRI Pharma Inc., Sendai 982-0804, Japan

**Keywords:** nicotine-induced conditioned place preference, nucleus accumbens, Ca^2+^/calmodulin-dependent protein kinase II, fatty acid-binding protein 3, dopamine D2 receptor

## Abstract

We previously demonstrated that fatty acid-binding protein 3 null (FABP3^−/−^) mice exhibit resistance to nicotine-induced conditioned place preference (CPP). Here, we confirm that the FABP3 inhibitor, MF1 ((4-(2-(1-(2-chlorophenyl)-5-phenyl-1H-pyrazol-3-yl)phenoxy) butanoic acid), successfully reduces nicotine-induced CPP scores in mice. MF1 (0.3 or 1.0 mg/kg) was orally administered 30 min before nicotine, and CPP scores were assessed in the conditioning, withdrawal, and relapse phases. MF1 treatment decreased CPP scores in a dose-dependent manner. Failure of CPP induction by MF1 (1.0 mg/kg, p.o.) was associated with the inhibition of both CaMKII and ERK activation in the nucleus accumbens (NAc) and hippocampal CA1 regions. MF1 treatment reduced nicotine-induced increases in phosphorylated CaMKII and cAMP-response element-binding protein (CREB)-positive cells. Importantly, the increase in dopamine D2 receptor (D2R) levels following chronic nicotine exposure was inhibited by MF1 treatment. Moreover, the quinpirole (QNP)-induced increase in the level of CaMKII and ERK phosphorylation was significantly inhibited by MF1 treatment of cultured NAc slices from wild type (WT) mice; however, QNP treatment had no effect on CaMKII and ERK phosphorylation levels in the NAc of D2R null mice. Taken together, these results show that MF1 treatment suppressed D2R/FABP3 signaling, thereby preventing nicotine-induced CPP induction. Hence, MF1 can be used as a novel drug to block addiction to nicotine and other drugs by inhibiting the dopaminergic system.

## 1. Introduction

Nicotine is a major reinforcer of tobacco and can easily induce compulsive use. Nicotine-induced addiction is difficult to quit and generally causes withdrawal symptoms following cessation of chronic exposure [1,2]. Nicotine, an extrinsic agonist of nicotinic acetylcholine receptors (nAChRs), exerts its physiological function in the ventral tegmental area (VTA) by binding pentameric ligand-gated ion channels, causing the opening of cation channels [3,4] and providing a water-filled channel through the membrane that facilitates the influx of calcium and sodium ions [5]. The heteromeric α4β2 nAChR and homomeric α7 nAChR are considered the main targets in nicotine-induced addiction because of their abundance and activation of the midbrain dopamine (DA) pathways [6,7]. The α4β2 nAChRs mediate the depolarization of dopaminergic (DAergic) neurons in the VTA [8] and are also the targets of some smoking-cessation treatments such as varenicline [9,10].

DA receptors belong to the G protein-coupled receptor (GPCR) superfamily. Numerous studies have reported that dopamine D1 receptors (D1Rs) are positively associated with nicotine-induced addiction. Nicotine stimulates DA release, contributing to the increase in cAMP and the influx of Ca^2+^ through D1R activation, resulting in extracellular signal-regulated kinase (ERK) phosphorylation [11]. Nicotine significantly increased DRD1 mRNA levels in the VTA, prefrontal cortex, and striatum [12]. Epidemiological investigations have revealed correlations between nicotine-induced addiction and five single nucleotide polymorphisms in the DRD1 gene, suggesting that DRD1 mediates nicotine addiction [13].

However, the critical roles of dopamine D2 receptors (D2Rs) in nicotine addiction remain controversial, although emerging evidence suggests that in addition to D1Rs, D2Rs are also involved in the abuse of drugs, including nicotine. D1-expressed medium spiny neurons (MSNs) are crucial in reward-based learning acquisition, and D2-expressed MSNs are required for conversion when the learning strategy is altered [14]. We previously used D2R null mice to demonstrate that D2Rs are crucial for nicotine-induced conditioned place preference (CPP) acquisition [15]. Solis et al. [16] reported that cocaine- and amphetamine-induced behavioral sensitization was also eliminated in D2R null mice. Robustly reduced reinforcement of the reward effects of drug abuse and impaired locomotor activity were also documented following a reduction in D2R availability [17,18,19]. Moreover, administration of a D2R antagonist blocked the nicotine-induced increase in conditioned reinforcement behaviors [20].

Ca^2+^/calmodulin-dependent protein kinase II (CaMKII) alpha, which binds to D2R via the D2R IL3 domain, is important for the regulation of intracellular signaling cascades such as ERK and CaM-dependent signaling [21,22]. Intracerebroventricular infusion of CaMKII antagonists blocked CPP establishment and CaMKII activation in the NAc and VTA of mice [23]. GluA1-Ser831 phosphorylation induced by CaMKII activation is necessary for nicotine-induced behavioral sensitization in rat caudate-putamen [24]. Furthermore, phosphorylation of ΔFosB Thr149 by CaMKIIα is enhanced by chronic cocaine use [25]. Mice with the CaMKIIα T286A mutation exhibited impaired cocaine-induced CPP behavior [26]. Meta-analysis revealed that the speed of acquisition of cocaine reinforcement is required for the CAMK2A gene [26]. Collectively, D1Rs, D2Rs and CaMKIIα positively regulate nicotine-induced addiction and the abuse of other substances.

Fatty acid-binding proteins (FABPs) function as cellular shuttling proteins of long-chain polyunsaturated fatty acids to their proper intracellular compartments because they are lipophilic and insoluble in water [27]. In mammals, FABPs include 12 family members that are expressed in different tissues and organs [28]. There are three main types of FABPs expressed in the central nervous system: FABP3, FABP5, and FABP7 [29]. FABP3 is associated with D2R [30] and specifically binds to the D2 long receptor at the insert region of 29 amino acid sequences (G242-V270) in the third cytoplasmic loop [31]. FABP3 null (FABP3^−/−^) mice exhibit reduced D2 receptor extrapyramidal motor functions [31], suggesting the essential role of FABP3 in D2R functions. Moreover, we reported that FABP3^−/−^ mice show failed CPP induction following chronic nicotine administration [32]. We recently successfully developed an FABP3 inhibitor, MF1 (4-(2-(1-(2-chlorophenyl)-5-phenyl-1H-pyrazol-3-yl) phenoxy) butanoic acid), which has a high affinity for FABP3 [33,34]. MF1 was previously reported to inhibit DAergic neuronal loss, α-synucleinopathy-related symptoms in 1-methyl-4-phenyl-1,2,3,6-tetrahydropyridine (MPTP)-treated and α-syn-preformed fibril infusion in mice [34,35], and α-syn oligomerization in FABP3-overexpressing neuro-2A cells [33]. However, the effect of MF1 on DAergic signaling and the underlying mechanism through which MF1 induces nicotine addiction remain unclear.

In this study, we demonstrated that MF1 treatment inhibits CPP acquisition in mice following chronic nicotine administration. We also analyzed the mechanism underlying CPP-induction failure through CaMKII and ERK activation. Quinpirole (QNP)-induced activation of CaMKII and ERK in cultured NAc slices was significantly ameliorated by MF1 treatment, whereas QNP and MF1 had no effect on cultured NAc slices from D2R null mice.

## 2. Results

### 2.1. Nicotine-Induced CPP in Mice Was Significantly Inhibited by MF1 Treatment

We previously reported that FABP3^−/−^ mice failed to acquire CPP following chronic nicotine administration [32]. Thus, we first evaluated the effect of the FABP3 inhibitor, MF1, on CPP behavioral tests in the presence of nicotine. The CPP apparatus used in this study is shown in Figure 1A. The chemical structure of MF1 and the experimental schedule of the CPP behavioral tasks are shown in Figure 1B and Figure 1C, respectively. Consistent with our previous studies, we observed that 14 consecutive days of nicotine administration significantly increased mice CPP scores in the conditioning, withdrawal, and relapse tests (conditioning: 171.8 ± 9.8; withdrawal: 181.0 ± 5.2; relapse: 181.4 ± 12.4, F (15, 78) = 9.663, *p* < 0.01 vs. preconditioning vehicle-treated group). However, pretreatment with a 0.3 mg/kg dose of MF1 prior to nicotine administration successfully blocked the acquisition of CPP in mice (conditioning: 120.3 ± 11.0; withdrawal: 118.9 ± 17.7; relapse: 119.4 ± 14.5, *p* < 0.01 vs. nicotine-treated group in each phase). A 1.0 mg/kg dose had a more potent effect on abrogating nicotine-induced CPP (conditioning: 94.4 ± 3.7; withdrawal: 78.7 ± 9.0; relapse: 93.9 ± 9.0, *p* < 0.01 vs. nicotine-treated group in each phase). The presence of MF1 alone did not affect CPP scores relative to vehicle-treated mice (Figure 1D). To evaluate the involvement of the DAergic system in nicotine-induced behavior, we verified whether blocking D1Rs and D2Rs using proper antagonists could affect the formation of nicotine-induced CPP. We found that treating mice with the D1R antagonist, SCH23390, or the D2R antagonist, eticlopride, was enough to block nicotine-induced CPP behaviors in the conditioning test (nicotine group: 151.4 ± 7.4; SCH23390 group: 96.1 ± 4.0; eticlopride group: 97.7 ± 6.6, F (5, 30) = 7.291, *p* < 0.01 vs. nicotine-treated group) (Figure 1E), suggesting that acquisition of nicotine-induced CPP requires the involvement of D1Rs and D2Rs.

### 2.2. MF1 Ameliorates Nicotine-Induced Kinase Phosphorylation and Reduces DA Receptor Levels

Next, we evaluated CaMKII and ERK phosphorylation levels in the NAc and hippocampal CA1 regions, since these kinases have previously been reported to play crucial roles in establishing nicotine-induced addiction [36,37]. Significant nicotine-induced CPP score increases were correlated with significantly increased CaMKII autophosphorylation and ERK phosphorylation levels in the NAc (pCaMKII, nicotine-treated group: 208.4 ± 21.8, F (2, 15) = 16.70, *p* < 0.01 vs. vehicle-treated group; pERK, nicotine-treated group: F (2, 15) = 5.798, *p* < 0.05, vs. vehicle-treated group). However, these increases were significantly inhibited by MF1 pretreatment (pCaMKII, MF1-pretreated nicotine group: 115.9 ± 8.4, *p* < 0.01 vs. nicotine-treated group; pERK, MF1-pretreated nicotine group: 97.6 ± 5.6, *p* < 0.05 vs. nicotine-treated group). Nicotine administration significantly elevated D1R levels in the NAc (D1R, nicotine-treated group: 223.5 ± 11.2, F (2, 15) = 15.74, *p* < 0.01, vs. vehicle-treated group), whereas this increase was blocked by pretreatment with MF1 (D1R, MF1-pretreated nicotine group: 131.9 ± 18.6, *p* < 0.01, vs. nicotine-treated group). D2R levels in the NAc were also significantly enhanced following chronic nicotine administration (D2R, nicotine-treated group: 139.5 ± 3.6, F (2, 15) = 30.61, *p* < 0.01 vs. vehicle-treated group). Notably, we observed that MF1 pretreatment completely blocked nicotine-induced increase in D2R levels in the NAc (D2R, MF1-pretreated nicotine group: 107.0 ± 4.3, *p* < 0.01 vs. nicotine-treated group). FABP3 levels were not affected by the administration of nicotine or MF1 (FABP3, nicotine-treated group: 94.5 ± 8.4, F (2, 15) = 0.1348, *p* > 0.05, vs. vehicle-treated group; FABP3, MF1-pretreated nicotine group: 102.1 ± 11.8, *p* > 0.05, vs. nicotine-treated group) (Figure 2A–G).

Similar results were observed in the analysis of the hippocampal CA1 region: nicotine exposure significantly increased CaMKII autophosphorylation and ERK phosphorylation levels (pCaMKII, nicotine-treated group: 167.6 ± 14.2, F (2, 15) = 11.41, *p* < 0.01 vs. vehicle-treated group; pERK, nicotine-treated group: 134.6 ± 2.2, F (2, 15) = 39.51, *p* < 0.01 vs. vehicle-treated group). These alterations were significantly ameliorated by MF1 pretreatment (pCaMKII, MF1-pretreated nicotine group: 128.4 ± 5.2, *p* < 0.05 vs. nicotine-treated group; pERK, MF1-pretreated nicotine group: 95.6 ± 2.7, *p* < 0.01 vs. nicotine-treated group). Moreover, nicotine exposure significantly increased D1R and D2R levels in the hippocampal CA1 region (D1R, nicotine-treated group: 121.5 ± 6.2, F (2, 15) = 8.451, *p* < 0.05, vs. vehicle-treated group; D2R, nicotine-treated group: 126.5 ± 3.4, F (2, 15) = 15.79, *p* < 0.01 vs. vehicle-treated group). However, the increase in nicotine-induced DA receptor levels was significantly inhibited by MF1 pretreatment (D1R, MF1-pretreated nicotine group: 93.4 ± 4.3, *p* < 0.01 vs. nicotine-treated group; D2R, MF1-pretreated nicotine group: 95.1 ± 5.4, *p* < 0.01 vs. nicotine-treated group). FABP3 levels were not affected by either nicotine or MF1 treatment (FABP3, nicotine-treated group: 101.9 ± 5.5, F (2, 15) = 0.0439, *p* > 0.05, vs. vehicle-treated group; FABP3, MF1-pretreated nicotine group: 102.1 ± 3.2, *p* > 0.05 vs. nicotine-treated group) (Figure 2H–N).

### 2.3. Inhibition of Nicotine-Induced Kinase Phosphorylation Is Associated with the Responsiveness of CREB/c-Fos Signals

We next focused on analyzing whether MF1 pretreatment affects CREB/c-Fos signals, since Ca^2+^ mediates transcription in the nucleus by regulating CREB and the proto-oncogene c-Fos [38]. The numbers of single- and double-positive cells in the NAc were counted using immunofluorescence staining. First, in accordance with the increased CaMKII autophosphorylation levels observed in immunoblotting assays, we observed that the number of autophosphorylated CaMKII-positive cells and autophosphorylated CaMKII/c-Fos double-positive cells were both significantly elevated following chronic nicotine administration to mouse NAc (pCaMKII-positive cells, nicotine-treated group: 1197.9 ± 30.4, F (2, 12) = 15.26, *p* < 0.01 vs. vehicle-treated group; pCaMKII/c-Fos double-positive cells, nicotine-treated group: 612.3 ± 57.7, F (2, 12) = 7.969, *p* < 0.05 vs. vehicle-treated group). These increases were successfully inhibited by MF1 pretreatment (pCaMKII-positive cells, MF1-pretreated nicotine group: 777.3 ± 95.0, *p* < 0.01 vs. nicotine-treated group; pCaMKII/c-Fos double-positive cells, MF1-pretreated nicotine group: 378.0 ± 44.0, *p* < 0.05 vs. nicotine-treated group) (Figure 3A–C). We also observed that the number of phosphorylated CREB-positive cells and the responsiveness of c-Fos were both upregulated following chronic nicotine administration (pCREB-positive cells, nicotine-treated group: 1257.8 ± 59.9, F (2, 10) = 37.21, *p* < 0.01 vs. vehicle-treated group; pCREB/c-Fos double-positive cells: 1018.3 ± 38.2, F (2, 10) = 59.14, *p* < 0.01 vs. vehicle-treated group), in line with the immunofluorescence results observed for autophosphorylated CaMKII and c-Fos. Chronic nicotine-induced increases were inhibited by MF1 pretreatment (pCREB-positive cells, MF1-pretreated nicotine group: 591.0 ± 67.6, *p* < 0.01 vs. nicotine-treated group; pCREB/c-Fos double-positive cells, MF1-pretreated nicotine group: 463.2 ± 26.1, *p* < 0.01, vs. nicotine-treated group) (Figure 3D–F). Overall, these results suggest that MF1 ameliorated chronic nicotine-induced behaviors by preventing CaMKII and CREB activities and responsiveness to c-Fos expression.

### 2.4. MF1 Treatment Rescues QNP-Induced Increase in Kinase Phosphorylation in Cultured NAc Slices from WT Mice

Next, to verify whether MF1 rescued the nicotine-induced increase in CaMKII and ERK activities and further attenuated the responsiveness of CREB/c-Fos signals by inhibiting D2R via impaired D2R/FABP3 signals, we cultured brain slices from WT and D2R null mice. The experimental schedule for brain slice preparation and culture is illustrated in Figure 4A. First, immunoblotting results of cultured NAc slices from WT mice showed that QNP stimulation significantly increased both CaMKII autophosphorylation and ERK phosphorylation levels (pCaMKII, QNP-treated group: 136.0 ± 11.5, F (2, 13) = 5.986, *p* < 0.05, vs. vehicle-treated group; pERK, QNP-treated group: F (2, 13) = 6.738, *p* < 0.05 vs. vehicle-treated group). QNP induced an increase in the levels of these kinases; however, these were significantly inhibited by MF1 pretreatment (pCaMKII, MF1-pretreated QNP group: 105.6 ± 2.9, *p* < 0.05 vs. QNP-treated group; pERK, MF1-pretreated QNP group: 107.7 ± 7.5, *p* < 0.05 vs. QNP-treated group) (Figure 4B–D). On the contrary, both QNP and MF1 treatment had no effect on the increase in the levels of these kinases in cultured NAc slices dissected from D2R null mice (pCaMKII, QNP-treated group: 94.2 ± 2.3, F (2, 13) = 1.826, *p* > 0.05 vs. vehicle-treated group; MF1-pretreated QNP group: 99.5 ± 2.7, *p* > 0.05 vs. QNP-treated group; pERK, QNP-treated group: 104.0 ± 3.4, F (2, 13) = 2.270, *p* > 0.05 vs. vehicle-treated group; MF1-pretreated QNP group: 107.0 ± 2.0, *p* > 0.05 vs. QNP-treated group) (Figure 4E–G). Next, we evaluated the effect of the D2R antagonist, eticlopride, on CaMKII autophosphorylation and ERK phosphorylation levels in cultured NAc slices from WT mice. Compared with vehicle-treated groups, we observed that CaMKII autophosphorylation and ERK phosphorylation levels were significantly decreased following eticlopride treatment (pCaMKII, ETIC-treated group: 78.0 ± 4.2, *p* < 0.01; vehicle-treated group; pERK, ETIC-treated group: 70.7 ± 3.8, *p* < 0.01, vs. vehicle-treated group) (Figure 4H–J). These results imply that MF1 pretreatment prevents CaMKII and ERK activities by abrogating D2Rs in mouse NAc.

### 2.5. Majority of GABAergic—But Not Cholinergic—Neurons Colocalize with FABP3 in the NAc

Finally, to determine whether FABP3 is also functionalized with γ-aminobutyric acidergic (GABAergic) and/or cholinergic neurons, we conducted double staining experiments to evaluate the possible colocalization of FABP3 using GAD67 and ChAT antibodies, since GABAergic and cholinergic neurons were reported to correlate with various D2R-related addictions [39,40]. Based on immunofluorescence staining data from NAc of WT mice, we observed that FABP3 predominantly colocalized with GAD67-positive cells but not with ChAT-positive cells. The quantitative results indicate that 76.5 ± 3.2% of GAD67-positive cells were colocalized with FABP3 (number of GAD67/FABP3 double-positive cells: 28.0 ± 3.2/mm^2^), while only 10.6 ± 3.3% of ChAT-positive cells were colocalized with FABP3 in the NAc (number of ChAT/FABP3 double-positive cells: 6.3 ± 1.9/mm^2^) (Figure 5A–D). Since the NAc receives DAergic innervation from the VTA, the nerve terminals of DAergic neurons stained with TH are shown in immunofluorescence images (Appendix A). Magnified images of double-positive immunofluorescence staining and reconstructive surface plots are presented in Figure 5E–H, which clearly demonstrate the colocalization of FABP3 with GAD67-positive cells but not with ChAT-positive cells in the NAc.

## 3. Discussion

In the present study, we report, for the first time, that MF1 treatment successfully prevents chronic nicotine-induced CPP acquisition in mice. We confirmed that this amelioration using MF1 was closely correlated with reduced CaMKII and ERK activities and reductions in DA receptor levels. Inhibition of the increase in CaMKII activity was correlated with CREB phosphorylation and c-Fos expression in mouse NAc. MF1 treatment significantly inhibited QNP-induced increase in CaMKII and ERK phosphorylation in cultured NAc slices from WT mice, whereas the treatment did not affect CaMKII and ERK phosphorylation in slices from D2R null mice. Taken together, the results show that MF1 prevented chronic nicotine-induced CPP by inhibiting D2R/FABP3 signaling and ameliorating CaMKII/CREB phosphorylation, thereby affecting c-Fos expression. This mechanism explains how MF1 treatment prevents chronic nicotine-induced CPP induction.

D2Rs have been correlated with addictive drug reinforcement and relapse [41]. Some studies have indicated that D2R protein levels in NAc of adolescents were significantly upregulated following exposure to nicotine, and cocaine-induced reinforcement was inhibited by blocking D2Rs [42]. Furthermore, both D1Rs and D2Rs are critical in the abuse of nicotine and other drugs. D1Rs and D2Rs mRNA levels in the NAc, which were analyzed using in situ hybridization and RNase protection assays, were both upregulated in rats exposed to tobacco smoke [43]. Clinical studies have indicated that higher levels of D1Rs in the caudate and D2/3Rs in the NAc are positively correlated with reward effects [44]. Consistent with these studies, we observed that injection of both the D1R antagonist, SCH23390, and the D2R antagonist, eticlopride, completely blocked chronic nicotine-induced CPP, suggesting that D1Rs and D2Rs are both involved in nicotine-induced CPP acquisition. Although morphine treatment increases the immunoreactivity of D1R, D2R, and DA transporter (DAT), reductions in D1R and DAT levels are associated with decreased CPP scores in rats [45]. Rats exercised under aerobic conditions were reported to have reduced levels of D1R-like receptors in the NAc and olfactory tubercle, but increased levels of D2R-like receptors in the caudate putamen in both sexes, which may be a possible reason for the attenuated drug-induced addictive behaviors [46]. Some investigations revealed that exposure to nicotine reduced D1R levels but did not affect D2R levels in the prefrontal cortex of adolescent rats [47]. These discrepancies are still not well understood, although one possible reason is that nicotine exerts different biological effects by activating D1R or D2R in a dose-dependent manner [48]. In the present study, we found that D1Rs and D2Rs were both necessary for chronic nicotine-induced CPP behaviors. Additionally, D1/D2 heterodimeric receptor signaling in the NAc and dorsal striatum is involved in substance abuse. The D1/D2 receptor heterodimer is expressed in a subset of MSNs that express both dynorphin and enkephalin in the NAc, the globus pallidus, and the caudate putamen [49,50]. The D1/D2 receptor heterodimer couples to the Gq protein to activate phospholipase C, which directly promotes DA, leading to intracellular Ca^2+^ release and CaMKII activation [51,52]. D1/D2 receptor heterodimers regulate glutamate transmission and synaptic plasticity by binding to CaMKIIα, glycogen synthase kinase 3, and brain-derived neurotrophic factor (BDNF) [53], thereby regulating cocaine- and amphetamine-induced addictions [54,55]. The number of striatal D1/D2 receptor heterodimers increases following amphetamine treatment [49]. Thus, to address these existing limitations, further investigations are required to reveal the role of the D1/D2 receptor heterodimer in nicotine addiction and to evaluate the pharmacological effect of MF1 on the D1/D2 receptor heterodimers.

D2R interacts with CaMKIIα via its third loop [22], and chronic levodopa treatment induces a significant increase in CaMKIIα-D2R interaction in striatal neurons [21]. QNP-induced D2LR stimulation activated the nuclear isoform of CaMKII in NG108-15 cells expressing D2LR [56]. Results from luciferase reporter gene assays showed that CaMKII and ERK positively regulate the D2R promoter, and that Zif268 is a potential transcription factor in the CaMKII-dependent pathway [57]. In accordance with these findings, we observed that an increase in D2R levels initiated by chronic nicotine exposure was accompanied by increased CaMKII autophosphorylation and ERK phosphorylation. Additionally, MF1 had an inhibitory effect on the phosphorylation of these kinases. We also demonstrated that the increase in CaMKII autophosphorylation levels induced by QNP stimulation was significantly inhibited by MF1 treatment, which was observed in cultured NAc slices from WT but not D2R null mice, suggesting that MF1 prevents CaMKII autophosphorylation by inhibiting D2Rs. FABP3 interacts and colocalizes with D2LR in vitro and in vivo [30,31,58,59]. Therefore, MF1, an FABP3 inhibitor, may also inhibit D2R/FABP3 function. In this study, we demonstrated that MF1 inhibits D2R/FABP3 function, thereby affecting DAergic signaling and subsequently influencing the Ca^2+^ signaling pathway to abolish chronic nicotine-induced CPP behaviors.

D2Rs are abundant in membrane rafts [60,61] and are internalized via the caveolae-mediated endocytic pathway [62,63]. The Gi-coupled receptor, D2LR, is internalized to activate intracellular ERK signaling following DA stimulation, and FABP3 and Rabex-5 proteins function in prolonged D2LR-mediated ERK signaling [64]. Moreover, the increase in Ca^2+^ signaling induced by QNP-stimulation of D2LR-transfected cells is correlated with nuclear CaMKII activation, and further enhances CaMKII-dependent BDNF levels, which indicates that D2L/Gα_i2_-coupled signaling likely stimulates nuclear Ca^2+^ signaling, including CREB/BDNF pathways [56]. Thus, internalized D2Rs stimulated by permeable agonists such as QNP, which are hydrophobic, can activate intracellular pooled D2LRs. Since FABP3 colocalizes with D2LR, and D2LR is internalized into intracellular areas, including the Golgi apparatus, to couple with FABP3, we speculated that MF1 did not only inhibit D2Rs located in plasma membrane rafts by inhibiting FABP3. In the present study, we observed that MF1 inhibited kinase phosphorylation following brain tissue extraction and immunoblotting analysis. Future studies are required to demonstrate how MF1 affects the internalization of D2Rs through FABP3 and CaMKII/ERK signaling in models of nicotine addiction.

## 4. Materials and Methods

### 4.1. Animals

Male C57BL/6J mice (8 weeks old) purchased from Clea Japan Inc. (Tokyo, Japan) and D2R null mice (8 weeks old) were used in this study. Mice were reared under conventional conditions, with constant temperature (23 ± 1 °C), humidity (55 ± 5%), and 12-h light/dark cycle. The mice were allowed access to food and water *ad libitum*. Four to seven of the mice were included in each group in all experiments. All experimental procedures involving animals were approved by the Committee on Animal Experiments of Tohoku University. Ethical approval for the study was obtained from the Institutional Animal Care and Use Committee of the Tohoku University Environmental and Safety Committee (2019PhLM0-021 and 2019PhA-024). We attempted to mitigate suffering in mice and used the minimum number of mice in all the experiments.

### 4.2. Chemical Agents

Nicotine hydrogen tartrate salt (0.5 mg/kg, i.p.) (Sigma-Aldrich, St. Louis, MO, USA) was dissolved in saline solution (0.9% NaCl) (Wako Pure Chemical, Osaka, Japan). The pH value was adjusted to 7.0 using dilute NaOH solution (Wako Pure Chemical, Osaka, Japan) prior to injection. The D1R antagonist, SCH-23390 hydrochloride (Sigma-Aldrich, St. Louis, MO, USA) (0.03 mg/kg, s.c.), was dissolved in saline. The D2R agonist, QNP (10 μM), was dissolved in artificial cerebrospinal fluid (aCSF) containing 124 mM NaCl, 3 mM KCl, 26 mM NaHCO_3_, 2 mM CaCl_2_·2H2O, 1 mM MgSO_4_·7H2O, 1.25 mM KH_2_PO_4_, and 10 mM D-glucose (Wako Pure Chemical, Osaka, Japan). The D2R antagonist, eticlopride hydrochloride (Sigma-Aldrich, St. Louis, MO, USA) (0.03 mg/kg, s.c., 20 μM ex vivo), was dissolved in saline and aCSF. MF1 (0.3, 1.0, and 10 μM, ex vivo) was synthesized as previously described [65] and prepared in a 0.5% carboxymethylcellulose sodium salt (CMC) solution and in 1% dimethyl sulfoxide (DMSO) in aCSF.

### 4.3. CPP Apparatus

The apparatus used to evaluate CPP behavior consisted of three rectangular compartments. Two of the compartments were designated as conditioning compartments, and differed in wall color and floor texture: One compartment was painted white and had a floor made of acrylic quadrangular sieve material. The other had entirely black walls and a floor with parallel stainless steel metal rods. The middle compartment was designated as the neutral compartment and had a gray floor and walls. Two guillotine doors were installed to separate the conditioning compartments.

### 4.4. CPP Task

#### 4.4.1. Acclimatization and Preconditioning Phase

The CPP task was conducted as previously described [32]. Five phases were included in the CPP task: acclimatization, preconditioning phase, conditioning phase, withdrawal, and relapse. Briefly, during acclimatization, each mouse was placed in the neutral compartment with the two guillotine doors open and allowed free access to all three compartments for 10 min. Acclimatization was conducted for four days and was aimed at allowing mice to adapt to the environment and to remove stresses. In the preconditioning test, each mouse was placed in the neutral compartment with the two guillotine doors closed for 5 min. The two guillotine doors were then removed and the mouse was allowed to explore all three compartments for 15 min. The retention time in each conditioning compartment was documented when a mouse had both forepaws in that compartment [66]. CPP scores were calculated as the ratios of the retention time in the conditioned compartment to the total time spent in the two conditioning compartments.

#### 4.4.2. Conditioning Phase

Environmental cues were introduced and established during the conditioning phase. Mice were injected with nicotine or saline (i.p.) and then kept in the related compartments with the two guillotine doors closed for 30 min each day for 14 consecutive days. MF1 (p.o.) and antagonists (s.c.) were administered 30 min prior to nicotine or saline injections. Ethanol was used to eliminate any odors during the interval between placements of mice. The CPP conditioning task was conducted as previously described.

#### 4.4.3. Nicotine Withdrawal and Relapse

Subsequently, in the withdrawal phase, each mouse received saline and/or MF1 treatment prior to placement in the related compartments for 5 days [67]. The CPP score for each mouse was documented in a nicotine-free state. Nicotine relapse was assessed the day after withdrawal. Each mouse received either nicotine or vehicle. Thereafter, each mouse was placed in the neutral compartment after 30 min and allowed to freely explore all three compartments for 15 min. CPP scores were calculated using the same method as for the preconditioning test.

### 4.5. Immunoblotting Analysis

Mice were sacrificed, brain tissues removed, and the NAc and hippocampal CA1 regions dissected. Brain tissues were frozen in liquid nitrogen and preserved at −80 °C until analysis. Immunoblotting was performed as previously described [68,69]. In brief, proteins were extracted from the NAc and hippocampal CA1 regions using homogenizing buffer (50 mM Tris-HCl, pH 7.4, 0.5% Triton X-100, 4 mM EGTA, 10 mM EDTA, 40 mM Na_2_P_2_O_7_·10H_2_O, 50 mM NaF, 1 mM Na_3_VO_4_, 1 mM dithiothreitol, 25 μg/mL pepstatin A, 50 μg/mL leupeptin, 50 μg/mL trypsin inhibitor, and 100 nM calyculin A). The insoluble pellet was removed by centrifuging for 10 min at 20,400× *g*. Proteins in the supernatants were measured and normalized using Bradford’s assay and then subsequently boiled in Laemmli sample buffer for 3 min. Equivalent amounts of proteins were loaded onto sodium dodecyl sulfate–polyacrylamide gel wells and transferred to polyvinylidene difluoride membranes after electrophoresis. TBST solution (50 mM Tris-HCl, pH 7.5, 150 mM NaCl, and 0.1% Tween 20) containing 5% fat-free milk powder was used to block the membranes. After several washes with TBST, membranes were incubated overnight at 4 °C with primary antibodies, including anti-phospho-CaMKII antibody (Thr286; 1:5000) [70], anti-CaMKII antibody (1:5000) [70], anti-phospho-ERK 1/2 antibody (Thr202/Tyr204; 1:1000; Cell Signaling, Woburn, MA, USA), anti-ERK 1/2 antibody (1:2000; Cell Signaling, Woburn, MA, USA), anti-D1R antibody (1:300; Santa Cruz Biotechnology, Dallas, TX, USA), anti-D2R antibody (1:300; Santa Cruz Biotechnology, Dallas, TX, USA), anti-FABP3 antibody (1:500; Proteintech, Rosemont, IL, USA), and anti-β-tubulin (1:5000; Sigma-Aldrich, St. Louis, MO, USA). The membranes were then incubated for 2 h at room temperature with appropriate secondary antibodies, including goat anti-mouse IgG (H + L)-HRP antibody (1:5000; Southern Biotech, Birmingham, AL, USA) or goat anti-rabbit IgG-HRP antibody (1:5000; Southern Biotech, Birmingham, AL, USA) following several washes with TBST. Protein blots were detected using an enhanced chemiluminescence immunoblotting detection system (Amersham Biosciences, NJ, USA) and X-ray film (Fuji Film, Tokyo, Japan). The intensities of protein bands were quantified using Image Gauge version 3.41 (Fuji Film, Tokyo, Japan).

### 4.6. Immunofluorescence

Immunofluorescence was performed as previously described [71,72]. Mice were anesthetized, perfused with ice-cold phosphate-buffered saline (PBS), and fixed with 4% paraformaldehyde (Sigma-Aldrich, St. Louis, MO, USA). Coronal brain slices (50-μm thickness) were prepared using a vibratome (DTK-1000, Dosaka EM Co. Ltd., Kyoto, Japan). After several washes with PBS, the brain sections were blocked with PBS containing 1% bovine serum albumin and 0.1% Triton X-100 for 1 h at room temperature and then incubated for 3 days at 4 °C with anti-phospho-CaMKII antibody (Thr286; 1:5000) [70], anti-phospho-cAMP response element-binding protein (CREB) (Ser133; 1:1000; Cell Signaling, Woburn, MA, USA), anti-c-Fos antibody (1:100; Santa Cruz Biotechnology, Dallas, TX, USA), anti-glutamate decarboxylase 67 (GAD67) antibody (1:1000; Millipore, Temecula, CA, USA), anti-choline acetyltransferase (ChAT) antibody (1:1000; Millipore, Temecula, CA, USA), anti-tyrosine hydroxylase (TH) antibody (1:1000; Millipore, Temecula, CA, USA), or anti-FABP3 antibody (1:50; Hycult Biotech, Uden, Netherlands). After washing thoroughly with PBS, the brain sections were incubated overnight at 4 °C with appropriate secondary antibodies, including Alexa Fluor 488-conjugated anti-mouse and/or anti-rabbit IgG antibody (1:500; Invitrogen, CA, USA). Subsequently, brain sections were washed with PBS and mounted in Vectashield (Vector Laboratories, Inc., Burlingame, CA, USA). A confocal laser scanning microscope (TCS SP8, Leica Microsystems, Wetzlar, Germany) was used for immunofluorescence imaging.

### 4.7. NAc Slices and Culture Preparations

Slices and cultures were prepared as previously described [73,74]. Briefly, mouse brains were rapidly removed, and transverse slices (450 μm thick) containing the NAc regions were prepared using a McIlwain tissue chopper. The slices were incubated in a perfusion chamber filled with persistently oxygenated (95% O_2_, 5% CO_2_) aCSF (described in the “Chemical Agents”) and warmed at 34 °C. The D2R agonist, QNP, and antagonist, eticlopride hydrochloride, were dissolved in aCSF. MF1 was dissolved in 1% DMSO prior to addition to aCSF. NAc regions were dissected from the cultured brain sections, frozen in liquid nitrogen, and stored at −80 °C for immunoblotting analysis.

### 4.8. Statistical Analysis

Data were analyzed using GraphPad Prism 8 (GraphPad Software, Inc., La Jolla, CA, USA) and results are presented as the mean ± standard error of the mean (SEMs). Two-group comparisons were conducted using Student’s *t*-test. Multigroup comparisons were conducted using one- or two-way analysis of variance (ANOVA) followed by Tukey’s post hoc test. Statistical significance was set at *p* < 0.05.

## 5. Conclusions

In summary, we demonstrated that MF1 treatment significantly inhibited chronic nicotine-induced CPP in mice. This inhibition was closely correlated with ameliorated D2R protein levels and CaMKII and ERK activation in the NAc and hippocampal CA1 regions. Elimination of the phosphorylation of these kinases was associated with CREB/c-Fos signals. MF1 treatment also significantly inhibited QNP-induced increases in CaMKII and ERK phosphorylation in cultured NAc slices from WT, but not D2R null, mice. Collectively, MF1 treatment inhibited D2R signaling through FABP3 and further prevented CaMKII activation and the corresponding increase in CREB and c-Fos levels, thus inhibiting chronic nicotine-induced CPP behavior acquisition in mice (Figure 6). Thus, the results of this study indicate that MF1 may be a novel candidate drug for preventing addiction to nicotine and other substances that affect the DAergic system.

## Figures and Tables

**Figure 1 ijms-24-06644-f001:**
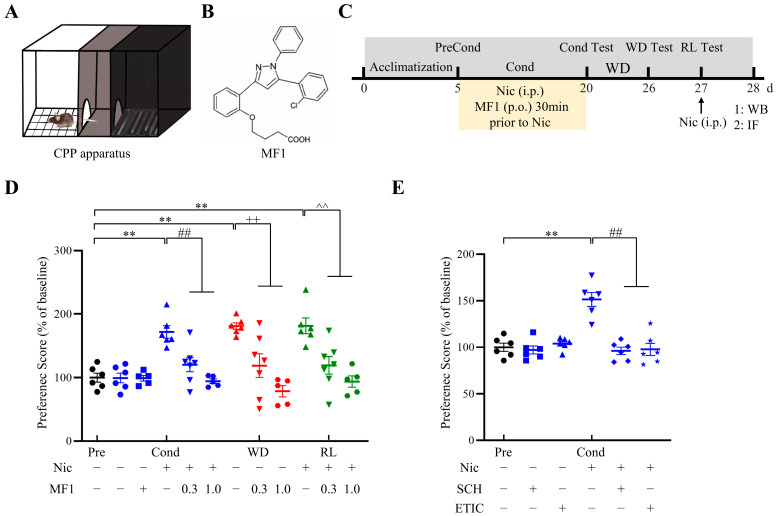
Experimental schedule and CPP tasks. (**A**) Graphical display of the CPP apparatus used in the present study. (**B**) Chemical structure of the FABP3 inhibitor, MF1. (**C**) Experimental procedure for CPP tasks and MF1 treatment. (**D**) Mice that received MF1 pretreatment showed significantly lower CPP scores in conditioning, withdrawal, and relapse phases. A 1.0 mg/kg dose showed more potent effects relative to a 0.3 mg/kg dose (*n* = 5–7 per group). ** *p* < 0.01 vs. preconditioning vehicle-treated group; ^##^ *p* < 0.01 vs. conditioning nicotine-treated group; ++ *p* < 0.01 vs. nicotine-treated group following withdrawal; ^^ *p* < 0.01 vs. nicotine-treated group following relapse. (**E**) Pretreatment with SCH23390 or eticlopride completely blocked nicotine-induced CPP in mice (*n* = 6 per group). ** *p* < 0.01 vs. preconditioning vehicle-treated group; ## *p* < 0.01 vs. conditioning nicotine-treated group. Precond, preconditioning test; Cond, conditioning; WD, withdrawal; RL, relapse; Nic, nicotine; WB, western blotting; IF, immunofluorescence staining; SCH, SCH23390 hydrochloride; ETIC, eticlopride hydrochloride. Error bars represent SEMs. The individual values in the CPP tasks are represented by data symbols, including solid dots, squares, triangles, inverted triangles, diamonds, and star pentagons, which are color-coded using black, blue, red, and green to distinguish between phases.

**Figure 2 ijms-24-06644-f002:**
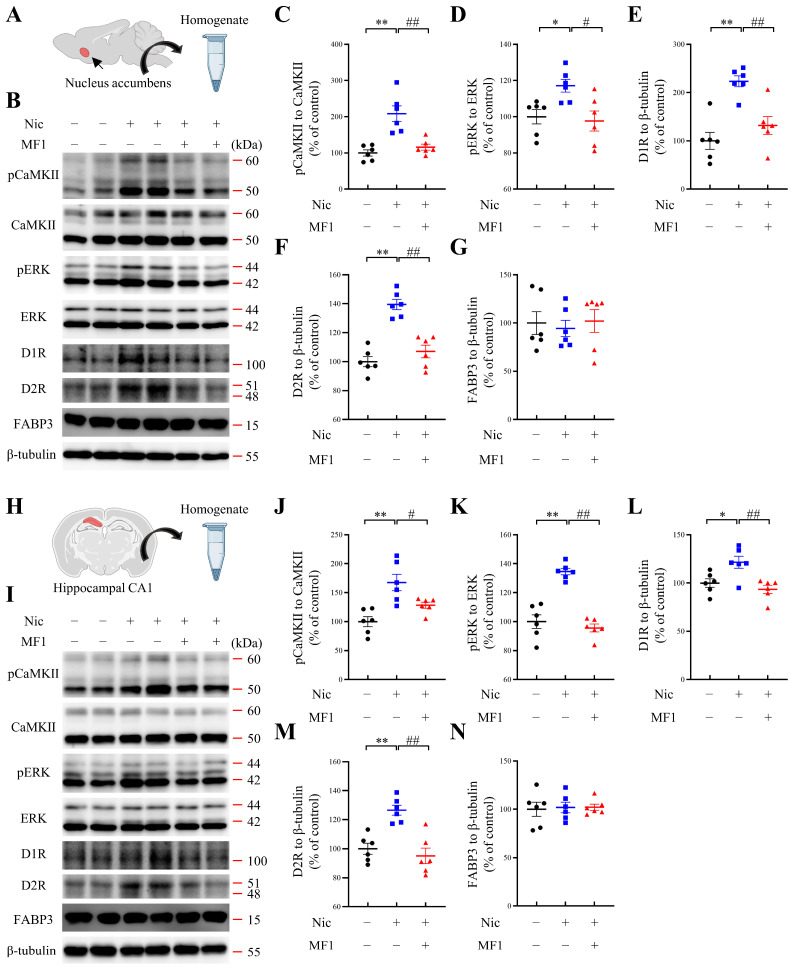
Kinase phosphorylation and DA receptor levels in CPP tasks. (**A**–**G**) Representative images and quantitative data for autophosphorylated CaMKII, CaMKII, phosphorylated ERK, ERK, D1R, D2R, FABP3, and β-tubulin in mouse NAc. Chronic nicotine exposure significantly increased levels of CaMKII autophosphorylation, ERK phosphorylation, D1R, and D2R, whereas these increases were clearly inhibited by pretreatment with MF1 (*n* = 6 per group). * *p* < 0.05 vs. vehicle-treated group; ** *p* < 0.01 vs. vehicle-treated group; # *p* < 0.05 vs. nicotine-treated group; ## *p* < 0.01 vs. nicotine-treated group. (**H**–**N**) Representative images and quantitative data for autophosphorylated CaMKII, CaMKII, phosphorylated ERK, ERK, D1R, D2R, FABP3, and β-tubulin in mouse hippocampal CA1 region. Consistent with results observed in the NAc, significantly elevated levels of autophosphorylated CaMKII, phosphorylated ERK, D1R, and D2R were blocked by pretreatment with MF1 (*n* = 6 per group). * *p* < 0.05 vs. vehicle-treated group; ** *p* < 0.01 vs. vehicle-treated group; # *p* < 0.05 vs. nicotine-treated group; ## *p* < 0.01 vs. nicotine-treated group. Nic, nicotine. Error bars represent SEMs. The immunoblotting analysis utilized data symbols, specifically solid dots, squares, and triangles, to represent individual ratio values, with each symbol color-coded using black, blue, or red to differentiate between different treatments within a group.

**Figure 3 ijms-24-06644-f003:**
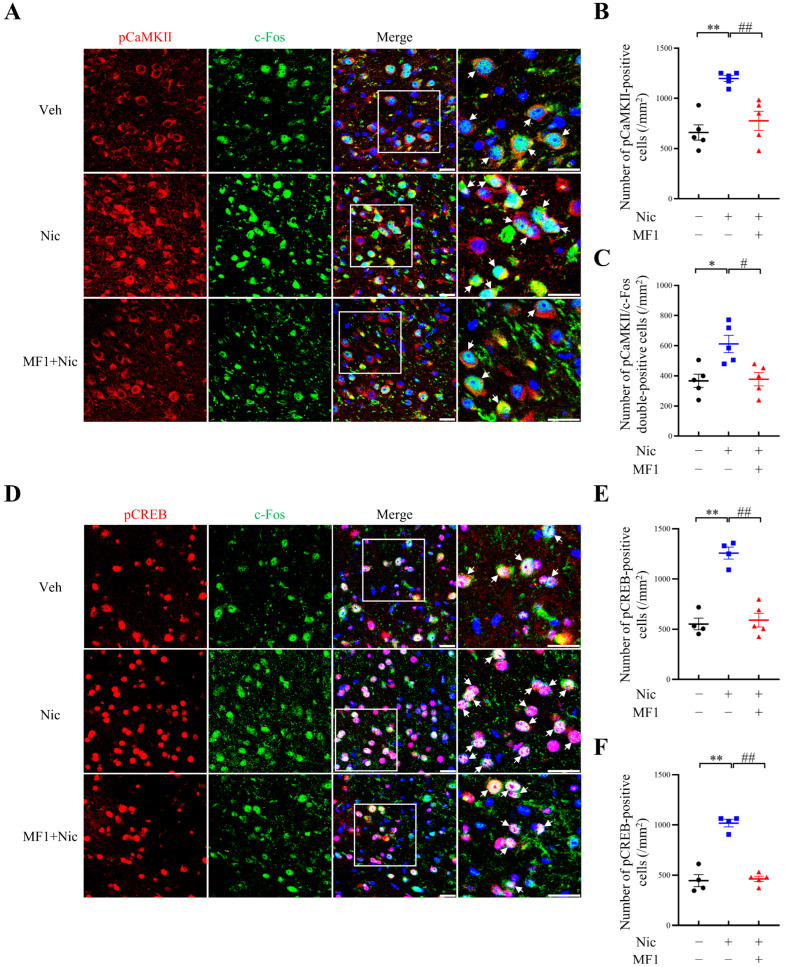
Numbers of phosphorylated CaMKII- and CREB-positive cells, and c-Fos expression in the NAc following CPP tasks. (**A**–**C**) Representative confocal images and quantitative data for autophosphorylated CaMKII-positive cells and autophosphorylated CaMKII/c-Fos double-positive cells in mouse NAc. Chronic nicotine administration significantly increased the numbers of autophosphorylated CaMKII-positive cells (red) and autophosphorylated CaMKII/c-Fos double-positive cells (green) in the CPP tasks, whereas these increases were restored by MF1 pretreatment (*n* = 5 per group). DAPI-stained nuclei are shown in blue. * *p* < 0.05 vs. vehicle-treated group; ** *p* < 0.01 vs. vehicle-treated group; # *p* < 0.05 vs. nicotine-treated group; ## *p* < 0.01 vs. nicotine-treated group. (**D**–**F**) Representative confocal images and quantitative data for phosphorylated CREB-positive cells (red) and phosphorylated CREB/c-Fos double-positive cells (green) in mouse NAc. Similar increases in the numbers of phosphorylated CREB-positive cells and phosphorylated CREB/c-Fos double-positive cells were observed following chronic nicotine exposure; however, these increases were inhibited in the MF1-pretreated nicotine group (*n* = 4–5 per group). DAPI-stained nuclei are shown in blue. ** *p* < 0.01 vs. vehicle-treated group; ## *p* < 0.01 vs. nicotine-treated group. Veh, vehicle; Nic, nicotine. Error bars represent SEMs. Scale bars, 25 μm. The immunofluorescence analysis utilized data symbols, specifically solid dots, squares, and triangles, to represent individual values, with each symbol color-coded using black, blue, or red to differentiate between different treatments within a group.

**Figure 4 ijms-24-06644-f004:**
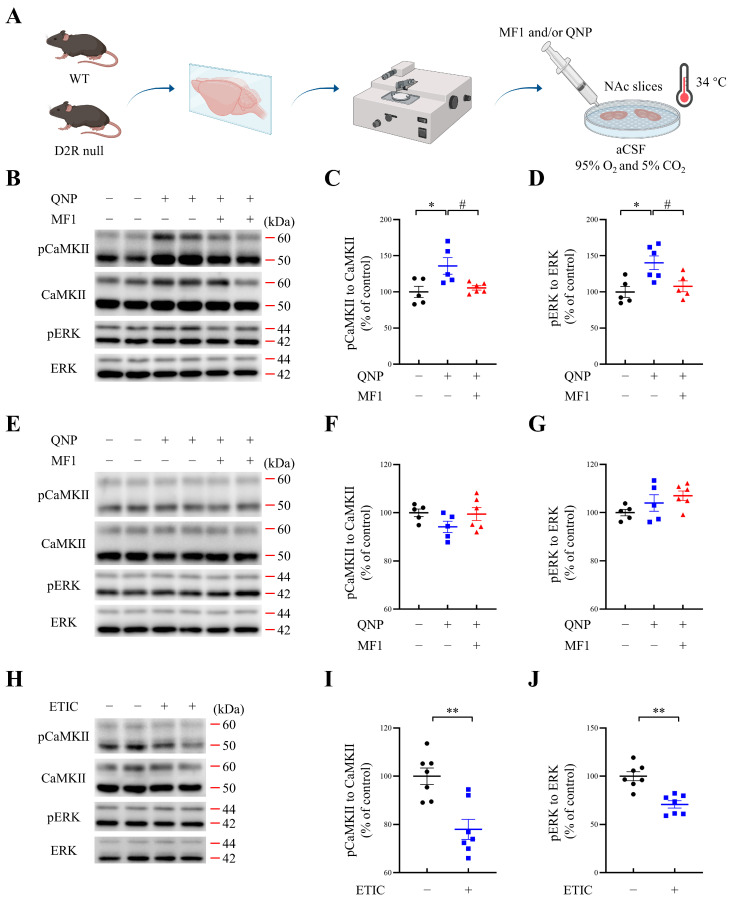
Kinase phosphorylation levels following MF1 treatment of cultured NAc slices from WT and D2R null mice. (**A**) Experimental illustration of mouse brain slice cultures. (**B**–**D**) Representative images and quantitative data for autophosphorylated CaMKII, CaMKII, phosphorylated ERK, and ERK in cultured NAc slices from WT mice. QNP stimulation significantly increased CaMKII autophosphorylation and ERK phosphorylation levels; however, these increases were inhibited by MF1 pretreatment (*n* = 5–6 per group). * *p* < 0.05 vs. vehicle-treated group; # *p* < 0.05 vs. QNP-treated group. (**E**–**G**) Representative images and quantitative data for autophosphorylated CaMKII, CaMKII, phosphorylated ERK, and ERK in cultured NAc slices from D2R null mice. Robust changes in levels of CaMKII autophosphorylation and ERK phosphorylation were not observed following QNP and MF1 pretreatment (*n* = 5–6 per group). (**H**–**J**) Representative images and quantitative data for autophosphorylated CaMKII, CaMKII, phosphorylated ERK, and ERK in cultured NAc slices from WT mice. Eticlopride treatment significantly reduced levels of CaMKII autophosphorylation and ERK phosphorylation relative to the vehicle-treated group (*n* = 7 per group). ** *p* < 0.01 vs. vehicle-treated group. QNP, quinpirole hydrochloride; ETIC, eticlopride hydrochloride; aCSF, artificial cerebrospinal fluid; WT, wild type. Error bars represent SEMs. The immunoblotting analysis utilized data symbols, specifically solid dots, squares, and triangles, to represent individual ratio values, with each symbol color-coded using black, blue, or red to differentiate between different treatments within a group.

**Figure 5 ijms-24-06644-f005:**
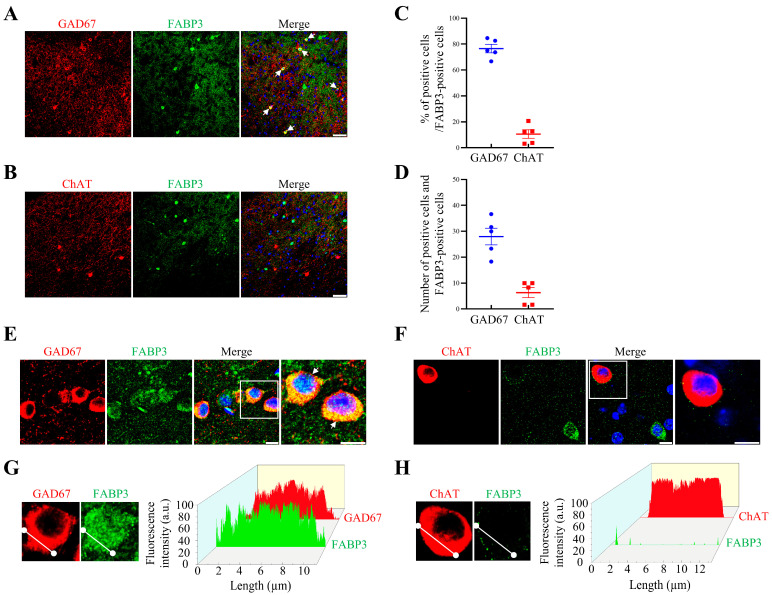
Colocalization of FABP3 in the NAc. (**A**–**D**) Representative images and quantitative percentage data of GAD67-positive cells (red) and ChAT-positive cells (red) colocalized with FABP3 (green), and the number of GAD67 and ChAT/FABP3 double-positive cells in the NAc. Most GAD67-positive cells—but not ChAT-positive cells—were observed to coexist with FABP3 (*n* = 5 per group). DAPI-stained nuclei are shown in blue. Error bars represent SEMs. Scale bars 100 μm. (**E**–**H**) Representative magnified images of GAD67, ChAT, and FABP3, and surface plots of fluorescence intensity, shown individually. Results are in agreement with the above-mentioned data and clearly indicate that most GAD67-positive cells, but not ChAT-positive cells, are colocalized with FABP3 in the NAc. Surface plots were measured and reconstructed using Fiji (Fiji Is Just ImageJ) (Version 1.53u) and OriginPro 2021b (9.85) software. a.u., arbitrary unit. Scale bars 10 μm. The immunofluorescence analysis utilized data symbols, specifically solid dots and squares, to represent individual values, with each symbol color-coded using blue or red to differentiate between different positive cells.

**Figure 6 ijms-24-06644-f006:**
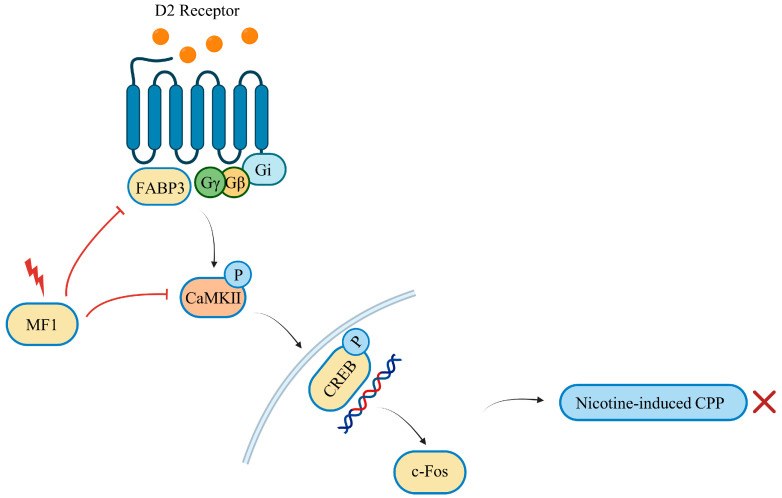
Schematic representation of the inhibition of nicotine-induced CPP behaviors in the NAc by MF1. Nicotine-induced activation of D2R/FABP3 signaling is inhibited by MF1 treatment, which subsequently prevents CaMKII and CREB activation and further reduces c-Fos transcriptional expression. The inhibition of CaMKII/CREB phosphorylation and the reduction in c-Fos expression are insufficient to elicit nicotine-induced CPP behaviors in mice.

## Data Availability

The data presented in this study are available upon request from the corresponding author.

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
