# Peer review of "Amelioration of Nicotine-Induced Conditioned Place Preference Behaviors in Mice by an FABP3 Inhibitor"

_ijms, 2023, doi:10.3390/ijms24076644_

Round 1
Reviewer 1 Report
The manuscript "Amelioration of Nicotine-Induced Conditioned Place Preference Behaviors by FABP3 Inhibitor in Mice" aimed to confirm that the FABP3 inhibitor: MF1, inhibits the nicotine-induced CPP scores in mice following chronic nicotine administration.
This work successfully demonstrated in both in vivo and ex vivo experiments that MF1 suppressed D2R/FABP3 signaling and prevented nicotine-induced CPP induction. They showed that MF1 has the potential to block nicotine addiction mediated by the dopaminergic system.
However, small details should be addressed i.e. the Y-axis orientiation in a few graphs including the Graphical Abstract.
Methods should also be showed in detail, for example, explaining how many animals per treatment were used.
Author Response
Reviewer #1:
The manuscript "Amelioration of Nicotine-Induced Conditioned Place Preference Behaviors by FABP3 Inhibitor in Mice" aimed to confirm that the FABP3 inhibitor: MF1, inhibits the nicotine-induced CPP scores in mice following chronic nicotine administration.
This work successfully demonstrated in both in vivo and ex vivo experiments that MF1 suppressed D2R/FABP3 signaling and prevented nicotine-induced CPP induction. They showed that MF1 has the potential to block nicotine addiction mediated by the dopaminergic system.
However, small details should be addressed i.e. the Y-axis orientation in a few graphs including the Graphical Abstract.
Ans: According to the comment, we revised the Graphical Abstract and other graphs.
Methods should also be showed in detail, for example, explaining how many animals per treatment were used.
Ans: Thank you very much for the comment. We added the description “Four to seven mice of above-mentioned ages were employed in each group in all experiments” in the “Materials and Methods” and added the animal number used in each figure legend.
Reviewer 2 Report
Following the analysis of the manuscript titled "Amelioration of Nicotine-Induced Conditioned Place Preference Behaviors by FABP3 Inhibitor in Mice", I appreciate the article's topic is interesting and I recommend that it should be revised taking into account the following observations:
- The abstract should follow the style of structured abstracts, but without headings, therefore rearrange it with more data for Materials and methods.
- Materials and Methods - provide the number and data for the study’s ethical approval. What was the number of animals used in the experiments?
- Discussion - clarify the strengths and the limitations of this study.
- Please provide an abbreviations list at the end of the manuscript.
- Update the references because many articles in the list have been published for more than 10 years, even 20 years. Both the Introduction and the Discussion should be based on presenting especially the latest evidence from the chosen topic.
Author Response
Reviewer #2:
Following the analysis of the manuscript titled "Amelioration of Nicotine-Induced Conditioned Place Preference Behaviors by FABP3 Inhibitor in Mice", I appreciate the article's topic is interesting and I recommend that it should be revised taking into account the following observations:
- The abstract should follow the style of structured abstracts, but without headings, therefore rearrange it with more data for Materials and methods.
Ans: According to the comment, we rewrote the abstract including materials and methods.
- Materials and Methods - provide the number and data for the study’s ethical approval. What was the number of animals used in the experiments?
Ans: We added the ethical approval and the number of animals used in each group in the “Materials and Methods” and figure legends. Four to seven mice of 8-week-old were employed in each group in all experiments conducted in this study.
- Discussion - clarify the strengths and the limitations of this study.
Ans: According to the comment, we revised the Discussion to clarify the strengths and limitations. We represented the strengths as follows.
“In the present study, we report for the first time that MF1 treatment successfully prevents chronic nicotine-induced CPP acquisition in mice. We confirmed that this amelioration was closely related to the reduced activities of CaMKII and ERK and the reduction of DA receptor levels. Elimination of CaMKII elevation was correlated with CREB phosphorylation and c-Fos expression in the mouse NAc. In cultured NAc slices, MF1 treatment significantly inhibited the QNP-induced increase in CaMKII and ERK phosphorylation in WT mice, whereas the treatment did not affect CaMKII and ERK phosphorylation in slices from D2R null mice. Taken together, MF1 prevented chronic nicotine-induced CPP by inhibiting D2R/FABP3 signaling and ameliorating CaMKII/CREB phosphorylation, thereby affecting c-Fos expression. This mechanism explains how MF1 treatment prevents chronic nicotine-induced CPP induction.” (Page 12, Line 297-307).
We also represented the limitations as follows.
“The D1/D2 receptor heterodimer is coupled to the Gq protein to activate phospholipase C, which directly promotes DA for intracellular Ca2+ release and the activation of CaMKII [51, 52]. D1/D2 receptor heterodimers regulate glutamate transmission and synaptic plasticity by connecting with CaMKIIα, glycogen synthase kinase 3, and brain-derived neurotrophic factor (BDNF) [53], thereby regulating cocaine- and amphetamine-induced addiction [54, 55]. The number of striatal D1/D2 receptor hetero-dimers increases following amphetamine treatment [49]. Thus, to address these existing limitations, our future investigations are required to reveal the properties of the D1/D2 receptor heterodimer in nicotine addiction and to evaluate the pharmacological effect of MF1 in terms of the D1/D2 receptor heterodimers.” (Page 12-13, Line 333-343).
- Please provide an abbreviations list at the end of the manuscript.
Ans: We made the lists into the manuscript.
- Update the references because many articles in the list have been published for more than 10 years, even 20 years. Both the Introduction and the Discussion should be based on presenting especially the latest evidence from the chosen topic.
Ans: According to the comment, we revised and added some latest references into our manuscript.
Round 2
Reviewer 2 Report
The manuscript has been improved but my last suggestion still remains.
Some references are way too old (e.g. 1993, 1995, 1997, 2000) and should be replaced with recent evidence on this topic.
Author Response
Reviewer #2:
The manuscript has been improved but my last suggestion still remains.
Some references are way too old (e.g. 1993, 1995, 1997, 2000) and should be replaced with recent evidence on this topic.
Ans: Thank you for the comments. According to the comment, we revised and replaced with the latest references (replaced references number: 1, 2, 3, 7, 17, 19, 27, 29) in revised manuscript.